# Macro- and Micronutrients from Traditional Food Plants Could Improve Nutrition and Reduce Non-Communicable Diseases of Islanders on Atolls in the South Pacific

**DOI:** 10.3390/plants9080942

**Published:** 2020-07-24

**Authors:** Graham Lyons, Geoff Dean, Routan Tongaiaba, Siosiua Halavatau, Kabuati Nakabuta, Matio Lonalona, Gibson Susumu

**Affiliations:** 1School of Agriculture, Food and Wine, University of Adelaide, Waite Campus, Urrbrae, South Australia 5064, Australia; 2Tasmanian Institute of Agriculture, University of Tasmania, Prospect, Tasmania 7250, Australia; geoffrey.dean@utas.edu.au; 3Agriculture and Livestock Division, Ministry of Environment, Lands and Agriculture Development, Tanaea, South Tarawa, Kiribati; rtongaiaba@gmail.com (R.T.); sao@melad.gov.ki (K.N.); 4UN FAO Intergovernmental Technical Panel on Soils, Tonga; halavatauj@gmail.com; 5Department of Agriculture, Ministry of Natural Resources, Energy and Environment, Vaiaku, Funafuti, Tuvalu; matiolnln@gmail.com; 6Gibson Susumu: Sustainable Agriculture Programme, The Pacific Community (SPC), Suva, Fiji; GibsonS@spc.int

**Keywords:** atolls, leafy vegetables, non-communicable diseases (NCD), nutrition security, mineral nutrients, natural biofortification

## Abstract

Pacific Islanders have paid dearly for abandoning traditional diets, with diabetes and other non-communicable diseases (NCD) widespread. Starchy root crops like sweet potato, taro, and cassava are difficult to grow on the potassium-deficient soils of atolls, and high energy, low nutrient imported foods and drinks are popular. Nutritious, leafy food plants adapted to alkaline, salty, coral soils could form part of a food system strategy to reduce NCD rates. This project targeted four atolls south of Tarawa, Kiribati, and was later extended to Tuvalu. Mineral levels in diverse, local leafy food plants were compared to reveal genotype–environment interactions. Food plants varied in ability to accumulate minerals in leaves and in tolerance of mineral-deficient soils. Awareness activities which included agriculture, health, and education officers targeted atoll communities. Agriculture staff grew planting material in nurseries and provided it to farmers. Rejuvenation of abandoned giant swamp taro pits to form diversified nutritious food gardens was encouraged. Factsheets promoted the most suitable species from 24 analyzed, with multiple samples of each. These included *Cnidoscolus aconitifolius* (chaya), *Pseuderanthemum whartonianum* (ofenga), *Polyscias scutellaria* (hedge panax), and *Portulaca oleracea* (purslane). The promoted plants have been shown in other studies to have anti-NCD effects. Inclusion of the findings in school curricula and practical application in the form of demonstration school food gardens, as well as increased uptake by farmers, are needed. Further research is needed on bioavailability of minerals in plants containing phytates and tannins.

## 1. Introduction

### 1.1. Epidemic of Non-Communicable Diseases (NCDs)

Since the 1940s the consumption of high-energy, low-nutrient foods, including white flour, sugar and polished rice by Pacific Islanders, combined with reduced exercise, has resulted in alarming rates of obesity, heart disease, diabetes, and certain cancers [1,2]. Indeed, around 70% of deaths in Pacific Island countries (PICs) are due to non-communicable diseases (NCDs) [3,4,5]. Apart from the tragic personal cost, premature death and disability undermines national economic productivity. These diseases occurred at very low rates when traditional diets and lifestyles predominated [6]. In addition, many PICs are affected by the “double burden” of NCDs and under-nutrition; for example, high rates of iron-deficiency anemia in Papua New Guinea, Fiji, Solomon Islands, and Tuvalu [7,8]. Pacific Islanders have paid dearly for forsaking traditional diets. A recent study emphasizes the need, during nutrition transitions, for public health initiatives to promote traditional diets high in vegetables, fruits, and lean protein and agricultural initiatives to promote farm diversity [9].

In addition to the health benefits of traditional diets, local food crop (including wild food) biodiversity strengthens the resilience of food systems to climate events through increasing crop species richness, thus improving food and nutrition security [10,11,12,13,14]. It is also economically advantageous. Growing foods such as leafy greens, breadfruit, pumpkin, and bananas to improve nutrition helps to reduce trade deficits associated with high consumption of imported foods in the Pacific. In Kiribati and Tuvalu (see Figure 1), imported food comprises about 65% of food eaten [15].

### 1.2. The Special Case of Atolls

Kiribati (population 114,000) and Tuvalu (11,000) are small Pacific nations where around half the people live on the main atolls of South Tarawa and Funafuti, respectively.

Atoll soils are formed almost entirely from coral (predominantly calcium carbonate with some magnesium). They are coarse textured with no clay, so water flows straight through them. Moreover, droughts are common in this part of the world. The soil is often salty, highly alkaline (pH (H_2_O) 8.6–9.2), and low in nutrients such as potassium, iron, and manganese, and, unless well composted, low in organic matter [16]. Furthermore, inorganic fertilizers and chemical pesticides are prohibited on many atolls as they could pollute valuable underground fresh water. Improving soil health through targeted composting, along with growing and eating nutritious crops on atolls should lead to improved diet, nutrition, and health.

### 1.3. Why Leafy Plants?

On many atolls, in particular those of the Southern Gilbert Islands (part of Kiribati, see Figure 1), which often experience drought, starchy root crops can be difficult to grow, resulting in low tuber/storage root yield. This is associated with potassium deficiency and lack of sufficient water during the weeks after planting. Potassium is needed to ensure adequate storage root initiation, and the high potassium content of tubers/storage roots depletes soil potassium with each harvest [17,18].

On the other hand, hardy leafy food plants can yield well under these conditions. Many different types of leafy vegetables and leaves of other plants/food crops are grown and eaten in the Pacific region (e.g., edible ferns, kangkong, amaranth, drumstick, and leaves of starchy root crops like taro, sweet potato, and cassava) [10,19,20,21,22,23]. When available, indigenous vegetables are usually inexpensive and thus affordable to most people in both urban and rural areas. Despite this, they are often overlooked and regarded as “low status foods” [1,2]. However, they are important for human health, being nutritious and rich in protein, minerals, vitamins (e.g., A, B, C, K), beneficial phytocompounds, and fiber [10,22,23,24,25,26,27,28,29,30,31,32]. A study in Africa found that “orphan” (unimproved) leafy vegetables were popular with farmers if they were full-season varieties with high leaf yield, and resistant to pests, diseases, and abiotic stress (e.g., drought, heat, salinity). Retailers and consumers valued good appearance, long shelf-life, affordability, and high nutritional value [33]. 

Iron provides an example of an important micronutrient found in leafy vegetables. Lack of iron can cause iron-deficiency anemia, common in women, inducing fatigue and weakness, and in children, affecting growth, energy levels, and learning ability. Purslane, pumpkin leaves, kangkong, yellow beach pea, and chaya are all good sources of iron [23,29].

Phytocompounds such as flavonoids, anthocyanins, polyphenols, and carotenoids are beneficial to humans as antioxidants and anti-inflammatory agents in reducing the risk of diabetes, heart disease, and cancers. Examples include glucosinolates in drumstick leaves and anthocyanins in purple sweet potato leaves. Certain carotenoids, notably beta- and alpha-carotene, are converted to vitamin A when eaten, especially if consumed with some oil (e.g., coconut cream) [34]. Others, notably lutein (which is usually abundant in leafy vegetables) and zeaxanthin are important for eye health and can reduce the risk of cataracts [35]. Importantly, given the current NCD pandemic, there is growing evidence for specific activity of certain plants against diabetes and cardiovascular disease, e.g., drumstick [36,37,38] and chaya [39,40].

### 1.4. Project Objective and Strategy

The objective of this project (2014–2019) was to support and enhance an awareness program aimed at increasing production of nutritious leafy plants to reduce rates of NCDs in Kiribati and Tuvalu. This was achieved through:A survey of mineral nutrients in local leafy food plants collected in Kiribati and Tuvalu.Assessment of plants for overall nutritional content, taste, and adaptation, in particular their tolerance of drought, soil salinity, and soil alkalinity, which characterize atolls, especially those of the Southern Gilbert Islands.Production of a factsheet series to promote the most suitable leafy food plants for atolls.Collaboration with the UN International Fund for Agricultural Development (IFAD)’s Outer Islands Food and Water Project, together with agriculture, health, and education officers in Kiribati and Tuvalu, to facilitate supply of planting material and participate in workshops and school talks.

## 2. Results and Discussion

### 2.1. G x E Study 

During the 2014 scoping study for the current project, we found 11 of the 12 leafy vegetables featured in the earlier factsheets growing on South Tarawa and Funafuti. This was surprising in view of the almost universal inhospitable coralline atoll soils, compared with, for example, soils of Solomon Islands. Most were growing in gardens and hedges close to homes; however, they were usually used for animal feed or as ornamentals. Clearly, raising awareness is an important program component, which includes school food gardens and curriculum development, farmer field schools, village workshops, and media promotion. See further discussion on this below.

Plant production is limited mostly by soil plant-available mineral content. Comparing mineral concentrations in genetically diverse plants provides insights into the plant–environment interactions that control mineral nutrient accumulation [41]. This not only enables identification of nutritious food plants for humans and animals, but also can lead to improvement in sustainable yield. 

Mineral concentrations in leaves varied widely with species, with five-fold variations common between species grown on the same soil (Table 1 and Table 2). Variation was less marked between sites (environment effect) for most minerals than is usually found, due to the relatively uniform coralline soils. For example, most minerals in leaves of chaya (*Cnidoscolus aconitifolius*) varied by no more than two-and-a-half-fold across seven sites, the exceptions being manganese and zinc (Table 3).

### 2.2. Natural Biofortifiers with Variability in Micronutrient Efficiency

Table 4 features the leafy plants which consistently accumulated the highest levels of minerals. These species could be described as natural biofortifiers of the corresponding nutrient. Leaves of pumpkin (*Cucurbita pepo*), purslane (*Portulaca oleracea*), and chilli (*Capsicum frutescens*) contained relatively high concentrations of most of the minerals, and thus, at least with respect to minerals, could be regarded as the most nutritious overall. Other studies also report the high nutritive value of these plants [42,43,44].

Species such as hedge panax and birdsnest fern were not observed with leaf chlorosis during this study, regardless of high soil pH. They are not exceptional Fe accumulators, e.g., birdsnest fern collected on Papaelise Island, Funafuti contained only 13 mg/kg DW of Fe, but looked healthy, and this compares with cassava, with a critical level for Fe of around 50 mg/kg [45]. Nevertheless, it is likely that plants such as birdsnest fern are efficient for Fe [46], and probably also for other nutrients, e.g., Mn, Cu, K in short supply in coralline soils. These plants seem to be able to function normally, especially with respect to photosynthesis, even when the nutrient is present at low plant-available levels in the soil. This is a different trait (involving different genes) to the ability to take up and accumulate high levels of a nutrient. Birdsnest fern is also an exceptional accumulator of K. Most cassava varieties, on the other hand, suffered from chlorosis, stunted growth, and lack of sizeable storage roots on the southern atolls. However, cassava and purslane are adept at extracting Zn from the soil and accumulating it in leaves.

### 2.3. Factsheets

In addition to the introductory factsheet, 12 species factsheets were produced, which feature the most atoll suitable nutritious leafy vegetables identified during the project. Several of these species have also been recognized for their nutritional value in other studies (Bailey, 1992; French, 2010; SPC, 2012). The featured plants are *Amaranthus viridis* (amaranth), *Cnidoscolus aconitifolius* (chaya, tree spinach), *Moringa oleifera* (drumstick tree), *Polyscias scutellaria* (hedge panax), *Pseuderanthemum whartonianum* (ofenga, Carruthers’ falseface), *Vigna marina* (yellow beach pea, beach cowpea), *Ipomoea aquatica* (kangkong), *Cucurbita pepo* (pumpkin), *Sechium edule* (choko), *Abelmoschus manihot* (bele, aibika, slippery cabbage), *Capsicum frutescens* (chilli), *Portulaca oleracea* (purslane, pigweed). Factsheet 13 discusses nutritional aspects of composting methods suitable for atolls.

Although not featured in the factsheets due to budgetary constraints (and their overall mineral levels were a little below the selected species) other nutritious leafy vegetables included *Asplenium nidus* (birdsnest fern), *Pisonia grandis* (big lettuce tree), and the leaves of these starchy root crops: *Ipomoea batatas* (sweet potato), *Manihot esculenta* (cassava), *Xanthosoma sagittifolium* (cocoyam), and *Colocasia esculenta* (taro). The nutritional value of these species has been noted in earlier studies [10,19,22,28]. 

Yellow beach pea was included more for its importance as a well-adapted legume on atolls than for its eating quality. It is an efficient N-fixer, with extensive root nodulation observed whenever sampled in Kiribati and Tuvalu, and is salt- and drought-tolerant. It grows better on strongly alkaline soils than *Mucuna*, *Pueraria, Centrosema, Gliricidia, Erythrina,* and *Sesbania*, can smother weeds, and its relatively high N, Fe, and Zn content make it a valuable green manure and compost component. Its seed pods are good to eat and highly nutritious when green, although all but the youngest of its leaves are chewy due to their high fiber content. A similar creeping legume is the Mauna Loa bean (*Canavalia cathartica*), which also thrives near beaches of Tuvalu and Kiribati, and has purple flowers, larger pods, and is a more vigorous tree-climber than *Vigna marina.*

### 2.4. Medicinal Effects

Although this project focused on the food/nutritional value of leafy green vegetables, traditionally in many countries they are also used for specific medical applications. For example, chaya (which originated in Mexico and Mesoamerica) protects the heart, liver, and kidneys from toxin damage [47,48]; drumstick (India and Pakistan) has anti-bacterial effects [49,50]; and bele (Papua New Guinea and Solomon Islands) is used for bone repair and treating osteoporosis [51]. Hedge panax, drumstick, chaya, bele, and purslane are galactogogues that can stimulate lactation [10,52,53,54]. Indeed, purslane’s generic name Portulaca means “to carry milk”. This plant is so ubiquitous and prolific globally that it is usually regarded as a weed. It is renowned for its high n-3 fatty acid content [14], and in this study was found to be the best accumulator of Mg, Fe and Zn of all the plants analyzed. Pumpkin thrives in composted atoll soils and is already grown widely in both countries. Ofenga, in particular the red-leaf form, is better known in Kiribati for its embalming ability than as food. 

High-protein species drumstick and chaya were fostered in Kiribati and Tuvalu under the Pacific Regional Agricultural Programme (PRAP) in the 1980s, and have adapted well to harsh atoll conditions [21]. Drumstick is high in *b*-carotene, sulphur, and selenium [14]. This species had the highest *b*-carotene level (427 mg/kg) of all plants analyzed in the earlier ACIAR project in the Pacific and northern Australia, and its protein is considered to be high in quality, with a similar pattern of essential amino acids as soybean [1]. It regularly accumulates around 12 times the concentration of selenium and around four times the concentration of sulphur compared to most other plants grown on the same soil. At the Vaiaku, Funafuti site (Table 1), drumstick leaves contained 25 times the Se concentration of the mean of the other plants growing there. Similar differences have also been observed in Africa [55]. This trait would be especially valuable in Sub-Saharan Africa, where these minerals are deficient in many soils [56,57,58]. Their deficiency is considered by some researchers to increase risk of HIV/AIDS [59].

Chaya is also renowned for its nutritional and medicinal effects. Like drumstick, it is an excellent source of high-quality plant protein and carotenoids and is renowned for its liver- and kidney-health enhancing effects [48]. 

Bele is not as climate- or insect-resilient as chaya, but has been included as its flavor is highly regarded, it is noted for health properties, including high levels of the important carotenoids lutein and *b*-carotene [1], and it grows well on composted soils with sufficient rainfall. It is the most popular leafy vegetable in Solomon Islands and Papua New Guinea. 

Especially important, given the high NCD (particularly diabetes) rates in the Pacific and northern Australia, are the anti-diabetes and anti-cardiovascular disease effects of most of the plants featured in the factsheets, demonstrated in scientific studies. Studies with evidence for this include the following: drumstick [36,37,38,50,60], amaranth [61,62,63], bele [64,65,66], chilli [67,68,69], purslane [70,71,72], kangkong [73,74,75], ofenga [76,77,78], hedge panax [79,80], chaya [39,40], pumpkin [81,82], and choko [83,84]. Their inclusion in the diet in sufficient quantity is likely to reduce the risk of diabetes and cardiovascular disease, not only by reducing glycemic load when they are included with high-carbohydrate meals, but also because of specific anti-diabetes effects.

### 2.5. Mineral Deficiencies of Atoll Food Plants

The plant leaf mineral data revealed that the most common mineral deficiencies in both countries were K and Mn. For example, 51% of the plant samples had K < 15,000 mg/kg, and 46% had Mn < 15 mg/kg. Phosphorus was mostly in the adequate range of 2500–4000 mg/kg, with 23% at marginal levels. Copper was marginal (<4 mg/kg) in 19% of leaf samples. Nitrogen deficiency was rare, and 33% of leaf samples had >4% N, testament to effective long-term composting, along with the inclusion of legumes, cassava, chaya, and drumstick (which are all inherently high in N) in the sample collection. Sites which had been composted for several years were higher in N (both nitrate and ammonium), available P, K, Mg, B, Cu, Fe, Mn, and Zn.

Iron deficiency, which is usually associated with alkaline soils, was not widespread, with most leaf samples > 30 mg/kg. Critical Fe levels are species-specific; as noted earlier, hedge panax can function normally with less Fe than can cassava. Likewise, most plants had sufficient Zn and B, with levels mostly in the range 30–70 mg/kg, and only 11% of leaf samples were <20 mg/kg in either.

Leaf Na levels were, unsurprisingly, relatively high, mostly >2000 mg/kg in Kiribati and >5000 mg/kg in Tuvalu, but symptoms of Na, Cl, or NaCl toxicity were not observed, even in cocoyam with leaf Na of 31,000 mg/kg on South Tarawa. High soil and plant levels of Ca and Mg counteract Na toxicity [85,86].

### 2.6. Giant Swamp Taro Food Garden

The value of home gardens comprising diverse, nutritious, traditional food crops to supplement the diet of subsistence households is well documented [1,11,87,88,89,90]. The cultivation of the giant swamp taro (*Cyrtosperma merkusii*, called babai in Kiribati and pulaka in Tuvalu) is traditional on atolls. Pits are dug by hand down to the water table, which in many Kiribati atolls is only 1–2 m below the surface. Many of these pits are now neglected but they provide a strong connection to both culture and fresh ground water.

In an adaptation of this system, kangkong can be grown in the water with the swamp taro. Hence the drought-tolerance requirement is waived for this species. Its vigorous growth means that, unless harvested regularly, it can intrude upon the giant swamp taro, and may need to be grown in a separate pit. The other crops can be grown on constructed terraces formed from the pit walls, and drumstick, ofenga, hedge panax, and beach cowpea are planted around the pit at ground level. Other crops, such as bananas, pawpaw, sweet potato, and annual vegetables can be included as well (Figure 2). These babai food gardens (BFG) (or pulaka food gardens (PFG) in Tuvalu) represent a mini food system that can, once established, contribute significantly to a family’s nutrition. For example, 100 g fresh weight of purslane leaves/stem will provide around 20% of an individual’s daily Zn requirement. The size of the garden can be as small as 40 m^2^ or as large as 0.3 ha. Even in crowded places, such as Betio on South Tarawa, there is usually room to plant a drumstick tree or two, which would soon provide a sustainable daily supply of leaves for a family.

On Nonouti, which has perhaps the toughest environment of the four atolls in the study, especially for drought frequency and duration, 27 BFG were commenced during the project, and at July 2019, 12 appeared well established and sustainable, a reasonable success rate for a new concept. Participating households usually have a BFG as well as food plants growing nearby on the land surface and around the house. These include sweet potato, ofenga, chaya, lemon grass, chilli, Brazilian spinach, pumpkin, banana, pawpaw, coconut, and breadfruit. 

### 2.7. How to Eat these Nutritious Vegetables

It is recommended to eat around three handfuls (around 100 g) of leafy vegetables each day. Some green leaves can be eaten uncooked, e.g., purslane, kangkong, and chilli, which preserves most vitamins, but it is usually preferable to cook them. It is important to wash the leaves in clean water first, to remove dirt and possible pathogenic microbes. Optimum cooking methods are steaming, simmering in a little water, baking, or stir frying in a little oil (ideally virgin coconut oil or coconut cream) for minimal time to limit nutrient loss. The cooking water should not be discarded, but instead used for soup. A simple method suitable for these vegetables is to chop them into small pieces (except drumstick, in which case strip the leaflets from the wiry petioles), simmer in water for 10–15 min, add coconut cream, and simmer for a further 10–15 min. Other ingredients can be added to enhance the flavor if desired. More elaborate recipes are included in several of the factsheets. A recipe book that includes most of the featured nutritious leafy plants was produced, written in Kiribati language, by the IFAD community awareness team. 

The bioavailability of Fe, Zn, and other minerals will be reduced by the presence of phytate, tannins, and polyphenols, e.g., in drumstick, chaya, and purslane [91,92,93,94]. The effects of these so-called antinutrients can be reduced by various cooking methods. In chaya, for example, boiling significantly reduced phytates, oxalates, and tannins [94] and virtually eliminated cyanogenic glycosides [93].

### 2.8. Awareness Program and Planting Material Provision

In order to achieve impact, the project collaborated with multiple government ministries (Agriculture, Health, Education, Works), churches, NGOs, Island Councils, and communities on the target atolls. In Kiribati, about 1500 farmers attended information and training sessions on growing, handling, cooking, and preserving locally grown foods. Our findings agreed with an FAO study in Samoa which found that the main external factors which influenced people’s decisions about food were availability, accessibility, cultural obligations, and family income [95]. 

Increasing awareness and generating interest must be met with availability of supply, whether planting material for home gardens or on a larger scale for farmers to produce for markets and tourism outlets [2]. Suitable planting material of the selected species was supplied via ALD nurseries on each atoll. The ALD Tanaea HQ, an infertile site with multi-micronutrient deficiencies, was transformed, using an improved watering system and composting, into an important germplasm source and model nutritious food garden. In Tuvalu, secondary bush was cleared, a water tank and irrigation system installed on Funafala island, Funafuti atoll and nutritious food crops grown to supply nearby Vaiaku, the main population center. This highlights the need for more resources to be devoted to conservation of diverse leafy food plants, starchy root crops, and fruits in the Pacific region [2,89].

The Pacific Community’s Centre for Pacific Crops and Trees (CePaCT), Suva plays a key role in germplasm conservation and distribution. Conservation of traditional crops is especially important in countries such as Papua New Guinea and Solomon Islands, where the natural environment is threatened by logging, mining, and oil palm establishment [89]. 

Value chain research is essential: the producer needs to be convinced that production of green leafy vegetables is worth the effort. Strategies are needed to deliver health benefits to consumers and economic benefits to local horticultural producers and other value chain participants [13]. 

Improving nutrition is usually seen as the task of health agencies, but it is apparent that a cross-sectoral and multi-agency food system approach is needed [96]. The NCD pandemic can be addressed by increasing diversity on the farm and extending this diversity (of which nutritious leafy vegetables form an important part) to the diet. Involvement of children in promotion of nutritious local foods is integral; in many countries their importance in influencing lifestyle factors, especially diet, is becoming recognized. For example, schools can include food gardens featuring the most nutritious local plants, provide more nutrition education, and students can transfer knowledge back to their villages [1]. In the current study, around 1700 students (to date) have attended awareness and training programs. Further research is needed on how to optimize awareness and promotion. This is crucial for Kiribati and Tuvalu, where leafy plants were not major components of traditional diets. Studies examining the bioavailability of minerals in these plants are also needed.

## 3. Materials and Methods

Adaptability of leafy food plants to tough atoll conditions was clear from observation on the four atolls and on South Tarawa and Funafuti, precluding the need for formal trials. A survey was conducted to identify the most nutritious leafy food plants, in terms of minerals and protein that grow in Kiribati and Tuvalu. Particular attention was paid to species that thrive in the atoll environment. Leaf tissue samples were collected in Kiribati and Tuvalu from 2014 to 2018 (*n* = 140), and with the inclusion of leaf mineral data from the previous Pacific-Northern Australia nutritious leafy vegetable project (ACIAR PC/2010/063) [1] (*n* = 274), a total of 414 samples (with 65 food plant species, of which 24 were found growing in Kiribati and Tuvalu, and also 50 species used for herbal medicines (18) or compost (32); usually multiple samples of the same species growing at different sites were analyzed) informed the factsheets produced during the current project. In Kiribati, samples were collected on the islands of South Tarawa, Abemama, Tabiteuea North, Nonouti, and Beru. In Tuvalu, samples were collected on Vaiaku, Funafala, and Papaelise.

As with the earlier project, an opportunistic genotype–environment (GxE) strategy was employed. This included sampling of single leafy vegetable species growing at different sites as well as sampling multiple species growing at the same site. Note that due to daily time and budgetary constraints, the data presented in Table 1, Table 2 and Table 3 (in Section 2) are for single samples. The locations reported in the tables were chosen to typify the leaf mineral concentrations found throughout the study for these species. 

This enabled the effects of environment (mostly soil type) and genetics (plant species/variety) to be separated, thus allowing an assessment of the ability of each species/variety to take up and accumulate essential minerals in their leaves. The minerals studied were the macronutrients nitrogen, phosphorus, potassium, calcium, magnesium, and sulphur, in addition to sodium, often present in “macro concentrations” but required in micro amounts; along with the micronutrients iron, manganese, boron, copper, and zinc. All of these macro- and micronutrients, with the possible exception of boron, are required by humans and animals as well as by plants. A sub-sample was analyzed for selenium, an essential micronutrient for humans and animals, but which is not required by higher plants. The analyses also enabled detection of any mineral deficiencies in the plants sampled.

Each leaf tissue sample comprised around two handfuls of relatively young leaves: not the youngest or older leaves, but, e.g., in sweet potato, the 5th to 9th youngest leaves, sampled from several representative plants, avoiding plants with disease (e.g., virus, scab) symptoms. The exception was pumpkin where just the tips (up to 25 cm) were sampled. If the plants were dusty (e.g., if the plants were growing near a road), they were washed in clean water. Samples were dried in a microwave oven or perspex covered trays soon after collection and placed in labelled plastic ziplock bags. Soil and compost samples were also collected from numerous sites in Kiribati and Tuvalu, and these will inform a future article.

The samples were brought to Australia under a Federal Department of Agriculture permit and irradiated. They were then acid digested and N analyzed by the combustion method (using an Elementar instrument, and limit of detection (LOD) calculated as 10 x the standard deviation of the calibration blank). Protein % was estimated by multiplying nitrogen % by 4.4. The other minerals listed above were analyzed by inductively coupled plasma atomic emission spectrometry (ICPOES) (using a radial CIROS instrument, and LOD as above), while Se was analyzed using ICP mass spectrometry (ICPMS) (using ICPMS method-*7B2G*, and LOD as above). Appropriate quality control measures were applied, including regular duplicate samples and analyses of aluminum and titanium to detect dust/soil contamination, which inflates Fe concentration.

In the previous study, carotenoids were also analyzed, and to minimize enzymatic degradation, samples were dried rapidly in a microwave oven as soon as practicable after collection, usually the same night. The carotenoids *beta*-carotene (the major pro-vitamin A carotenoid), lutein (usually the most abundant carotenoid in leaves), and *alpha*-carotene were analyzed by size-exclusion high-performance liquid chromatography (HPLC) at the Mares laboratory, Waite Campus, University of Adelaide [1]. Budgetary constraints precluded carotenoid analyses in the current study; however, these methodological details are included as carotenoid data are included for several of the species featured in the factsheets, the main study output.

The criteria for atoll suitable leafy plants, as listed under Project objective above, were (1) highly nutritious, (2) taste good, (3) tolerant of alkalinity (i.e., soil pH (H_2_O) > 8.5), (4) tolerant of salt and drought (with the exception of *Ipomoea aquatica*, kangkong, which grows in fresh water), and (5) easy to grow, prepare, and cook. 

For the factsheets, photographs, characteristics, uses, availability, propagation and growing methods, disease and pest threats, and advice on harvesting and storage were included for each species. Leaf mineral and carotenoid data (if the species was included in the previous factsheets) were presented in a table which included the featured leafy vegetable sampled at a representative site, compared with other leafy vegetables growing at the same site. English cabbage was also included, as a moderately nutritious yardstick, using average values of samples purchased at markets in the South Pacific. 

Factsheets 1, 2, 4, 5, 6, 8, 9, 10, and 11 were adapted from those published during the ACIAR PC/2010/063 project, Feasibility study on increasing the consumption of nutritionally-rich leafy vegetables by indigenous communities in Samoa, Solomon Islands and Northern Australia (www.remoteindigenousgardens.net>2013/08>new-resources-fact-sheets) [23].

The factsheets (500 sets) were graphically designed, printed, and laminated to improve durability by SPC, Suva, Fiji and distributed in Kiribati and Tuvalu, and also published online: www.researchgate.net>publication>327261351_Tackling_NCDs_from_the_ground_up_Nutritious_ leafy_vegetables_to_improve_nutrition_security_on_atolls [29]. Later, they were translated into the Kiribati and Tuvalu languages. A recipe book featuring the selected plants was compiled and distributed to communities.

The project also trialed and promoted starchy root crops (taro, sweet potato, cassava) and non-leafy vegetables, including beans, tomato, cucumber, capsicum, eggplant, and watermelon, and these components are reported elsewhere. 

## Figures and Tables

**Figure 1 plants-09-00942-f001:**
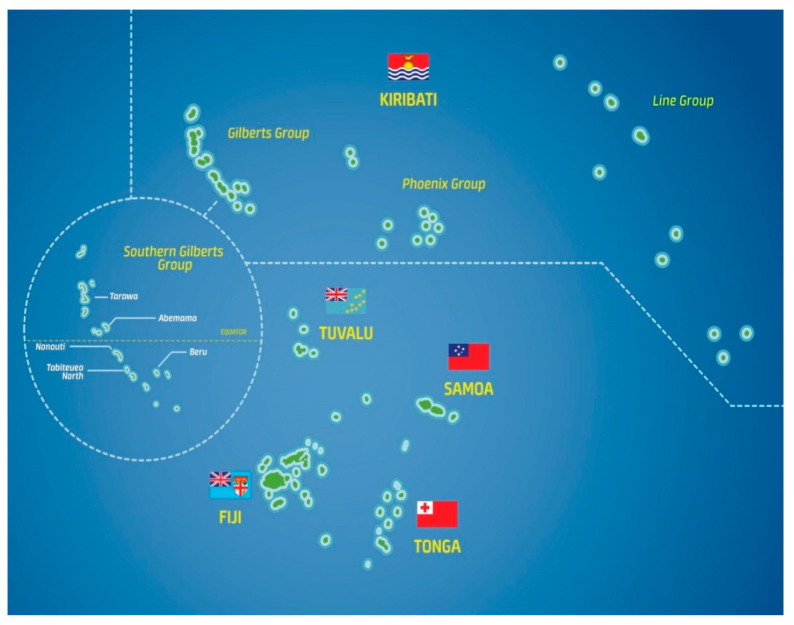
Map of part of the south-western Pacific Ocean, featuring Kiribati, Tuvalu, and Fiji, and (inset) the Southern Gilbert Islands. This project focused on Abemama, Tabiteuea North, Nonouti, and Beru, and (for value-chain activity) Abaiang, just north of Tarawa.

**Figure 2 plants-09-00942-f002:**
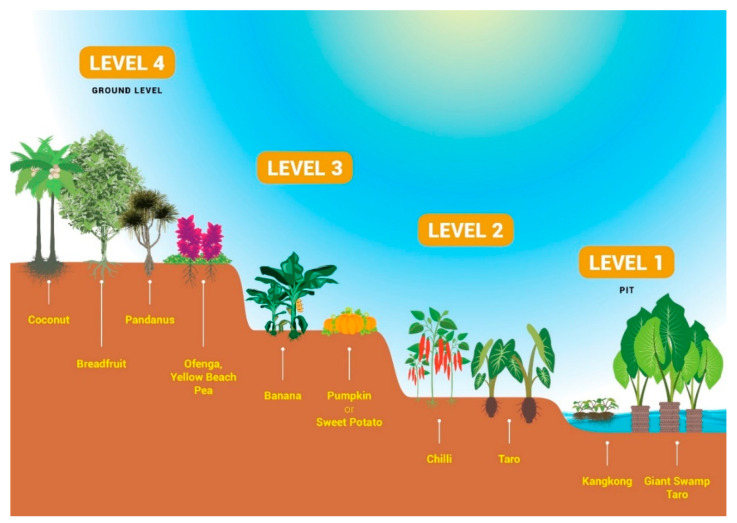
Layout of a babai/pulaka food garden. Other nutritious food crops can be substituted or added if desired.

**Table 1 plants-09-00942-t001:** Concentrations of macro- and micronutrients in leaves (dry weight basis) of different food plant species grown together on the same soil type at Vaiaku, Funafuti, Tuvalu in August 2014.

Species				Nutrient				
	Fe	Mn	B	Cu	Zn	Mg	K	N
	mg/kg	mg/kg	mg/kg	mg/kg	mg/kg	%	%	%
Brazil spinach (*Alternanthera sissoo*)	30	9	33	11	97	1.48	3.6	3.5
Chaya (*Cnidoscolus aconitifolius*)	76	19	19	9	42	0.55	1.64	5.1
Drumstick tree (*Moringa oleifera*)	52	12	21	7	39	0.61	1.09	5.2
Hedge panax (*Polyscias scutellaria*)	47	29	26	7	71	0.58	3.2	2.9
Lettuce tree (*Pisonia grandis*)	50	29	43	21	20	0.61	2.1	4.2
Ofenga (*Pseuderanthemum whartonianum*)	45	21	25	28	62	1.72	3.4	3.0
Purslane (*Portulaca oleracea*)	68	5	50	14	103	2.2	3.1	3.3
Variation (-fold)	2.5	5.8	2.8	5.6	5.2	4.0	3.3	1.8

Notes: Concentrations are on a dry weight (DW) basis throughout the manuscript. N % × 4.4 provides an estimate of crude protein %. Ca was uniformly high (range 1.61–2.20%). S was moderate in six species (0.21–0.38%) but high in drumstick tree (1.13%). Note that the data in Table 1, Table 2 and Table 3 are analyses conducted on single representative sub-samples of pooled samples for each species.

**Table 2 plants-09-00942-t002:** Concentrations of macro- and micronutrients in leaves (dry weight basis) of different food plant species grown together on the same soil type at Tanaea, South Tarawa, Kiribati in August 2014.

Species				Nutrient				
	Fe	Mn	Cu	Zn	Ca	Mg	S	N
	mg/kg	mg/kg	mg/kg	mg/kg	%	%	%	%
Cassava (*Manihot esculenta*)	35	37	10	88	1.30	0.67	0.34	5.0
Chilli (*Capsicum frutescens*)	32	25	8	63	3.89	1.80	0.65	3.3
Drumstick tree (*Moringa oleifera*)	65	20	5	32	1.58	0.74	1.16	5.4
Lettuce tree (*Pisonia grandis*)	42	31	17	16	2.34	1.00	0.32	3.3
Ofenga (green) (*Pseuderanthemum whartonianum*)	26	24	7	33	2.20	2.70	0.31	2.1
Ofenga (red)	30	22	10	25	1.72	1.30	0.24	3.2
Taro (*Colocasia esculenta*)	34	35	12	29	3.30	0.63	0.21	3.8
Variation (-fold)	2.5	1.9	3.4	5.5	3.0	4.3	5.5	2.6

**Table 3 plants-09-00942-t003:** Variation in selected minerals in leaves of *Cnidoscolus aconitifolius* (chaya) growing at seven locations in Tuvalu (sites 1 and 2) and Kiribati (sites 3–7) from 2014 to 2017. This study illustrates variation due mostly to differences in plant-available levels of these nutrients in soil. Most minerals (Zn, Mg, N) varied by less than three-fold.

Site		Nutrient		
	Mn	Zn	Mg	N
	mg/kg	mg/kg	%	%
1	17	50	0.50	5.0
2	4	43	0.90	4.9
3	12	35	0.56	5.2
4	19	42	0.55	5.1
5	10	27	0.71	4.7
6	4	50	0.56	5.9
7	32	79	1.11	4.2
Mean	14	47	0.70	5.0
Variation (-fold)	8.0	2.9	2.2	1.4

**Table 4 plants-09-00942-t004:** Selected mineral nutrients and the leafy vegetable species found (using opportunistic GxE analysis) in this study to be the most effective accumulators of these minerals in leaves. Samples were collected from various locations in Kiribati and Tuvalu. The values in brackets are representative concentrations of the relevant mineral for each species in this region.

Nutrient (Units)	Best Accumulators (Concentration in Leaf)
Iron (mg/kg)	Purslane (79), yellow beach pea (72), pumpkin (69), kangkong (68), chaya (65)
Manganese (mg/kg)	Giant swamp taro (94), cassava (34), taro (34), chilli (27)
Boron (mg/kg)	Chilli (60), drumstick (48), birdsnest fern (41), sweet potato (41)
Copper (mg/kg)	Tree lettuce (21), pumpkin (13), chilli (12), ofenga (11)
Zinc (mg/kg)	Purslane (119), cassava (107), pumpkin (97), hedge panax (81)
Calcium (%)	Chilli (3.8), bele (3.4), ofenga (2.7), hedge panax (2.5)
Magnesium	Purslane (2.5), ofenga (2.0)
Potassium (%)	Pumpkin (4.3), birdsnest fern (4.1), taro (3.0), kangkong (2.9)
Phosphorus (%)	Pumpkin (0.74), cassava (0.54), sweet potato (0.52)
Sulphur (%)	Drumstick (1.1), chilli (0.6), sweet potato (0.55)
Nitrogen (%)	Pumpkin (5.1), cassava (5.0), chaya (5.0), drumstick (4.7)
Selenium (µg/kg)	Drumstick (400)
Multiple nutrients	Pumpkin, purslane, chilli

Notes: Selenium is a micronutrient for humans and animals but not for higher plants; μg = micrograms; N % × 4.4 provides an estimate of crude protein % in leaves.

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
