# Peer review of "Macro- and Micronutrients from Traditional Food Plants Could Improve Nutrition and Reduce Non-Communicable Diseases of Islanders on Atolls in the South Pacific"

_plants, 2020, doi:10.3390/plants9080942_

Round 1
Reviewer 1 Report
I consider the presented research very important in the current context of an increasing human population in a world with less and less water and soil. However, the authors need to restructure the manuscript in order to make it detailed for the reader. Materials and methods need a deep reorganization as well as Results and Discussion.

Author Response
Dear Reviewer 1,
Thank you for you comments, in particular your view that the research is very important. It is always good to receive positive feedback.
Re your request to "restructure the ms to make it detailed for the reader", we have made some changes (ms with track changes attached), including fixing several typos and including some more analytical, etc details, but I would not say that these constitute a restructuring, and certainly not the "deep reorganisation" of the M&M and R&D that you have requested. Three of us are very experienced scientists, and we spent, on and off, 8 months putting this ms together. We request that you supply more detailed instructions in order for us to contemplate such major changes to a paper that we consider to be essentially suitable.
Please note that we have altered the title of the ms, making it more detailed and removing "biofortification", which some potential readers may associate with GM.

Reviewer 2 Report
This study focus on the relation of plant mineral status and human health. It is an interesting study, although the approach is essentially experimental and lacks a true scientific hypothesis. The authors compare several crops regarding their mineral composition in leaves, and the location effect (7 sites) on a species (Chaya). Most of the discussion section relies on previous results (research and surveys project) and qualitative assessment. Sometimes the paper resembles a review article. However, due to the importance of singularity and fragility of this ecosystem and due to the very specific soil conditions of these region, the data presented here deserves publication in this journal. The English style is very good, and the text is clear and well written.
Some specific comments:
I suggest changing the captions of table 1 and 2. In fact, tables refer to “Concentrations of macro and micronutrients determined in different species grown on the same soil type”. Authors may include other parameters such as CV, standard deviation to provide a more complete sample characterization.
In line 241, authors refer to soil P. It was not included in MM section. Please provide a reference.
There are a large pool of information in MM section which have not expressions in the paper (e.g. carotenoids and others). Please rewrite this section and present/discuss only the parameters that were presented in tables. I understand that this paper is a part of a major and important project but MM section should be in accordance to the results. If possible, provide also a short description of soil analysis in this section: pH, P and K, organic matter, etc.
Author Response
Dear Reviewer 2,
Thank you for your constructive comments and suggestions on our ms, in particular mention of the importance of the project and consideration that the data deserve to be published in Plants. It is always pleasing to receive positive feedback.
Latest draft with track changes is attached. We have altered the captions of Tables 1 and 2, according to your suggestion, and added notes (and also in the M&M) explaining the "single sub-sample of a representative sample at a representative site" method used here to illustrate our opportunistic GxE method.
We have removed mention of soil P in the Results. In fact we collected numerous soil and compost samples during this study, but this report focuses on plant minerals, while a later report will focus on minerals in soils and composts in Kiribati and Tuvalu. We have added mention of this in the M&M and added pH and mention of low OM to the brief soil details in the Introduction (ca line 70).
Thank you for the suggestion re possible extraneous material in the M&M. However, we would prefer the carotenoid details to remain, as several of the factsheets (which are the main project output) include carotenoid data from our previous study, while the tables in the current study comprise just a subset of the data used to formulate the factsheets. They are included largely to illustrate the opportunistic GxE method used in the study.
In addition, we altered the title to provide more detail and remove "biofortification", which some potential readers may associate with GM.

Round 2
Reviewer 1 Report
Dear author
Attached you can find some comments of the word document that you sent.
Two very important parameters for the evaluation of the significance of results are their repeatability and reproducibility. Results as those presented on tables 1-3 obtained from a single sub-sample of a composite sample have no statistical meaning. I encourage you to justify why did you prepare a composite sample with the plant material grown on different islands and performed a single analysis? If you knew, from other studies, that the location had not effect, why did use material from different locations? But if you expected a variation among the locations, why did you mix the material? Additionally, how repeatable and reproducible are the values reported?
I would like you to justify why you did not implement the suggestions given during the first round of revisions. I still do not see the point to have a chapter “how to cook …”.
There is also some confusion concerning the number of species (leafy species, medicinal...), factsheets and the origin of the data.

Author Response
Dear Reviewer 1,
Thank you for your further comments.
Re single samples analysed for Tables 1-3, as noted in the first revision (lines 373-375), we had neither the time to collect nor the budget to analyse multiple (say, 3) replications of each species at each site. As it was, I was usually up till ca 2am washing, microwave-drying, packaging and labelling the leaf samples after a day's collecting. In 2014, during a pilot study for this project, I found that there was little variation in concentration of leaf minerals in multiple samples from the same species at one location. This is largely due to the care taken to collect leaves of a standardized age-range. And this is especially so for these atolls, where the coralline soils are relatively uniform (e.g. compared with Solomon Islands). Nevertheless, there are still some differences in plant-available minerals between soils at different sites, especially for Mn Table 3). I don't think I have stated anywhere that location had no (or little) effect. To have collected and analyzed, say 3 replications of each species at each site would have resulted in a reduction in the number of sites sampled and greatly increased the analytical cost, with little or no, and probably less information gained. Statistical purity may not thus be served, but I stand by the validity and usefulness of the data presented. The information provided in the factsheets is informed by samples collected not only in Kiribati and Tuvalu in the years 2014-2018, but by my studies on Pacific Islands from 2009. I suggest that any researcher who seeks to study thoroughly minerals in edible leafy plants growing on Pacific atolls will obtain similar results.
We made some changes earlier, at your suggestion, e.g. providing more analytical details. Reviewer 2 actually wanted less information in the M&M...I strove to achieve a balance. As I recall, you recommended "deep and profound" structural changes to both the M&M and R&D, and I replied with a request for more details on this. Please bear in mind that this paper, if published, will be open-access, and we hope it will be read, not just by academics/researchers, but by agricultural extensionists, public health nutritionists, teachers, church and community groups and smallholders in Pacific Island countries and beyond. Hence I would prefer that the section "How to cook..." remains. In my experience this is the most common question asked by women in the villages we visit to promote these plants.
I have clarified the numbers of plant categories collected/analyzed in the M&M, lines 370-378.
Reviewer 2 Report
The authors had improved this last version. No further commenys to the ms.
Author Response
Dear Reviewer,
Many thanks for your assistance with our paper,
The authors